# Generation of *App* knock-in mice reveals deletion mutations protective against Alzheimer's disease-like pathology

Kenichi Nagata[1], Mika Takahashi[1], Yukio Matsuba[1], Fumi Okuyama-Uchimura[1], Kaori Sato[1], Shoko Hashimoto[1], Takashi Saito [1,2] & Takaomi C. Saido [1]

Although, a number of pathogenic mutations have been found for Alzheimer's disease (AD), only one protective mutation has been identified so far in humans. Here we identify possible protective deletion mutations in the 3′-UTR of the amyloid precursor protein (*App*) gene in mice. We use an *App* knock-in mouse model carrying a humanized Aβ sequence and three AD mutations in the endogenous *App* gene. Genome editing of the model zygotes using multiple combinations of CRISPR/Cas9 tools produces genetically mosaic animals with various *App* 3′-UTR deletions. Depending on the editing efficiency, the 3′-UTR disruption mitigates the Aβ pathology development through transcriptional and translational regulation of APP expression. Notably, an *App* knock-in mouse with a 34-bp deletion in a 52-bp regulatory element adjacent to the stop codon shows a substantial reduction in Aβ pathology. Further functional characterization of the identified element should provide deeper understanding of the pathogenic mechanisms of AD.

[1] Laboratory for Proteolytic Neuroscience, RIKEN Center for Brain Science, Saitama 351-0198, Japan. [2] Department of Neuroscience and Pathobiology, Research Institute of Environmental Medicine, Nagoya University, Nagoya 464-8601, Japan. Correspondence and requests for materials should be addressed to K.N. (email: kenichi.nagata@riken.jp) or to T.C.S. (email: takaomi.saido@riken.jp)

Alzheimer's disease (AD) is a progressive neurodegenerative disorder, which is pathologically characterized by deposition of amyloid-beta peptide (Aβ), as well as hyperphosphorylated tau aggregation. Aβ is a 38–43 amino acid peptide produced from amyloid precursor protein (APP) through processing by two different proteases, β-secretase and γ-secretase. Genetic analyses of familial AD (FAD) provide fundamental evidence that Aβ accumulation plays a central role in AD pathogenesis[1]. Over 300 pathogenic mutations in the *App* gene or *presenilin* gene, encoding a catalytic subunit of γ-secretase, have been found to cause FAD through increased Aβ production and/or a change in the ratio of aggregation-prone Aβ species[2]. In contrast to the pathogenic mutations, only one protective mutation, which slows or prevents the onset or progression of AD pathology, has been identified to date. A previous human genomic study with a set of whole-genome sequence data from 1,795 Icelanders revealed that a missense mutation (p.A673T) on the *App* gene has a protective effect against the onset of AD, possibly via reducing the levels of amyloidogenic peptides[3,4]. However, subsequent replication studies failed to reproduce the protective effects of the variant due to its rarity[5,6], underscoring the difficulty of exploring protective mutations by random screening based on a large human dataset. To facilitate the identification of protective mutations that occur infrequently or do not occur naturally in the human population, targeted screening based on relevant animal models of AD is necessary.

We recently established single-*App* knock-in model mice carrying a humanized Aβ sequence, as well as two or three clinically causative mutations in the endogenous murine *App* gene[7,8]. Generating these mice, we unexpectedly found that *App* knock-in mice lacking the last two introns (intron 16, 17) and 3′-UTR did not display Aβ deposition or deleterious effects at any ages due to a substantial reduction in APP expression at both the transcriptional and translational levels (Supplementary Fig. 1). The results led us to predict that these regions in the *App* gene affect Aβ accumulation via alteration of APP expression.

To test this theory, we here focused our attention on the *App* 3′-UTR because the sequence is highly conserved between mice and humans. We performed CRISPR/Cas9-mediated genome editing in *App* knock-in mouse zygotes in order to directly delete the *App* 3′-UTR. Subsequent quantitative Aβ measurements, as well as APP expression analyses in the 6-month-old edited model mice provided that deletion of the *App* 3′-UTR mitigated Aβ accumulation in the brain through the reduction of APP expression levels. Further experiments using another combination of genome editing tools enabled us to narrow down a possible responsible element in the *App* 3′-UTR. Our results suggest that targeted screening strategy based on relevant AD models would be useful for the identification of novel protective AD mutations.

## Results

**Generation of the *App* knock-in mice with *App* 3′-UTR deletion.** First, we explored whether disruption of the endogenous *App* 3′-UTR prevents Aβ accumulation in our AD mouse model. To this end, we deleted ~700-bp of the 891-bp *App* 3′-UTR in the mouse model zygotes using CRISPR/Cas9 technology (Fig. 1a, b)[9,10]. We selected an *App* knock-in mouse line (NL-G-F)[7] as a suitable model for the following reasons: the genome editing technology cannot be applied to the conventional APP-Tg mice because they overexpress APP in a multicopy manner and because they fail to possess *App* non-coding regions that regulate transcription and translation of the gene. We designed a pair of CRISPR/Cas9 tools to delete the *App* locus between the two target sites (Fig. 1a). After checking the editing efficiency in culture cells, the CRISPR/Cas9 tools were injected into NL-G-F mouse zygotes (Supplementary Table 2). Two weeks after the birth of mice, we performed PCR-based genotyping using tail genomic DNA. Simultaneous genome editing revealed that cleaved fragments of the 3′-UTR could be detected in 14 founder mice (named as NL-G-F ΔUTR) out of 49 live-born pups (Supplementary Table 2). To exclude the possibility of heterogeneous genotypes among the tissues in the founder mice, as shown in previous studies[11,12], we

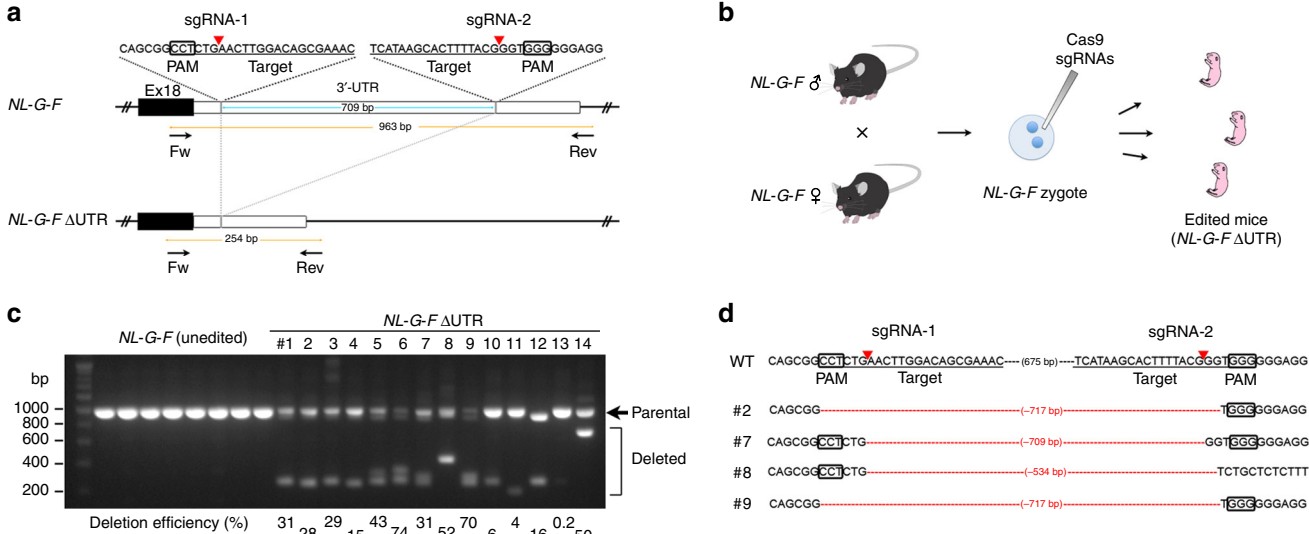

**Fig. 1** Disruption of *App* 3′-UTR in *NL-G-F* knock-in mice. **a** Positions of targeting sgRNAs and detection primers used for CRISPR/Cas9-mediated deletion of *App* 3′-UTR. Red arrowheads indicate Cas9 cleavage sites within the sgRNA target sites. **b** Strategy for genome editing of homozygous *NL-G-F* mouse zygotes. **c** PCR-based genotyping results of *NL-G-F* ΔUTR mice. Genotyping was performed using mouse brain samples. In contrast to unedited *NL-G-F*, *NL-G-F* ΔUTR showed various deletion in the *App* 3′-UTR. The deletion efficiency was calculated as the relative ratio of the intensity of deleted fragments (parenthesis) to that of parental band (arrow). The efficiency of each *NL-G-F* ΔUTR mouse is represented on the bottom of each lane. **d** Exact sequences of cleaved fragments from four representative *NL-G-F* ΔUTR mice (#2, #7, #8, and #9 from Fig. 1c). PCR products from mouse brain DNA samples were subcloned into the pTAC-1 vector and individual clones were picked and sequenced

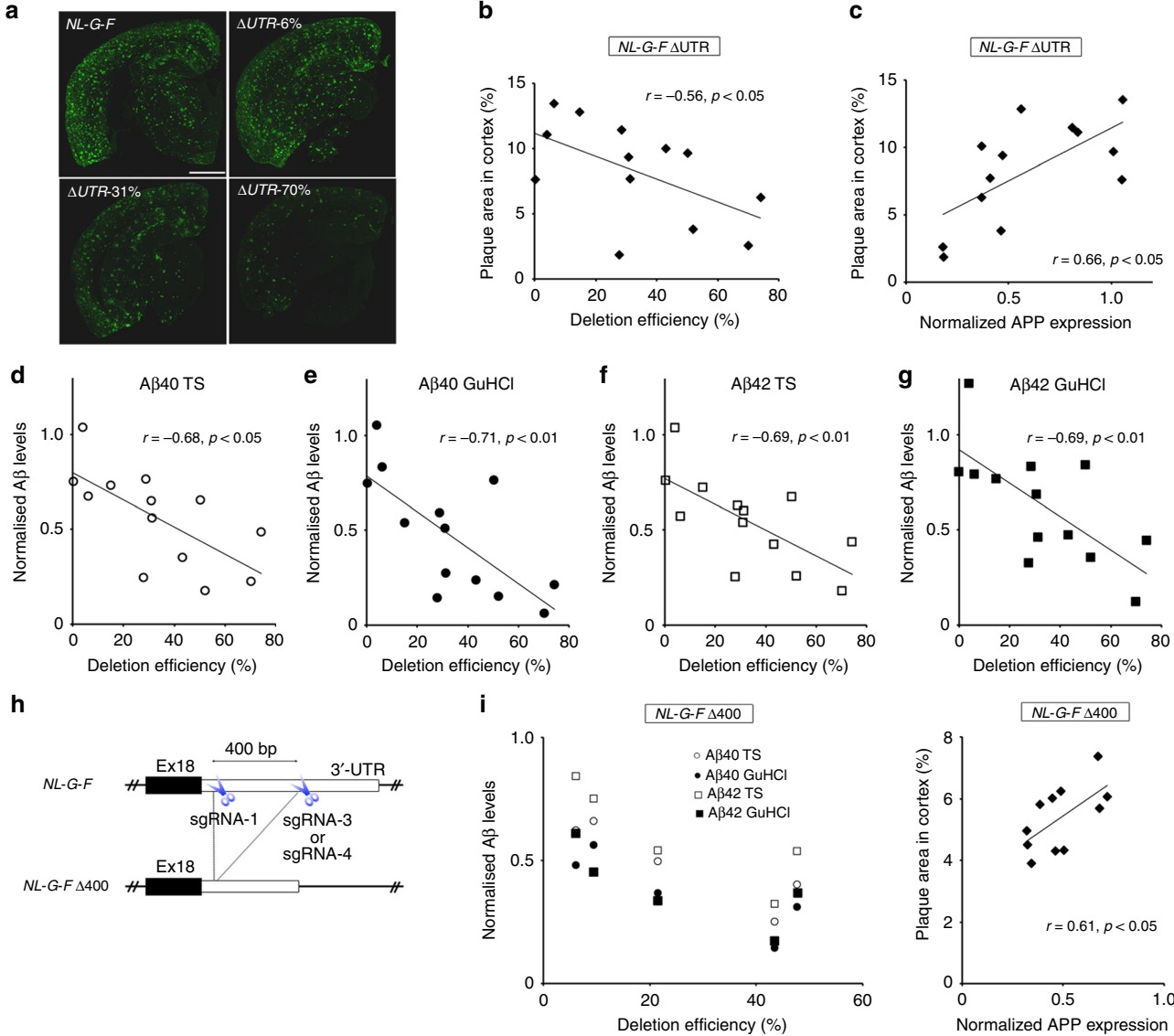

**Fig. 2** Prevention of Aβ accumulation in 6-month-old *App* 3′-UTR deleted *NL-G-F* mice. **a** Aβ pathology in 6-month-old *NL-G-F* ΔUTR mouse brains, as well as unedited control *NL-G-F* brains. Scale bar: 1 mm. **b** Negative correlation between Aβ accumulation and genome editing efficiency in *NL-G-F* ΔUTR mouse brains ($n = 13$). **c** Positive correlation between normalized APP protein expression and genome editing efficiency in *NL-G-F* ΔUTR mouse brains ($n = 13$). Western blotting was performed using an antibody against N terminus of APP protein. **d**–**g** Negative correlation between normalized Aβ levels and deletion efficiency in 6-month-old *NL-G-F* ΔUTR mouse brains ($n = 13$). Aβ$_{40}$ and Aβ$_{42}$ levels in both the Tris-HCl-buffered saline (TS) and GuHCl fractions were quantified by ELISA. **h** Strategy of CRISPR/Cas9-mediated *NL-G-F* Δ400. Positions of targeting sgRNAs are shown by scissors. **i** Negative correlation between Aβ levels and deletion efficiency in female *NL-G-F* Δ400 mouse brains ($n = 5$). **j** Positive correlation between normalized APP protein expression and genome editing efficiency in *NL-G-F* Δ400 mouse brains ($n = 11$)

also performed PCR-based genotyping using *NL-G-F* ΔUTR mouse brains. Deleted fragments could be distinguished from parental band by agarose electrophoresis. Based on the intensity of the deleted fragments, we calculated the deletion efficiency of each sample (Fig. 1c; 0.2–74%). The deletion efficiency correlated well between the tail and brain samples. The exact sequences of deleted fragments were then identified using Sanger sequencing (Fig. 1d). We excluded one mouse (#12) with coding sequence deletion from the founder candidates. In parallel, we performed off-target analyses using Sanger sequencing on seven potential sites in the 14 *NL-G-F* ΔUTR, and could not find any off-target mutations in the founder mice (Supplementary Table 3).

**Deletion of *App* 3′-UTR mitigated Aβ pathology in the *App* knock-in mice**. Next, we explored whether 3′-UTR deletion of

*App* has inhibitory effects on Aβ accumulation. To this end, we performed immunohistochemical analyses of 6-month-old *NL-G-F* ΔUTR brains with anti-Aβ antibody, and found a drastic reduction of Aβ accumulation in several samples with highly disrupted *App* 3′-UTR (Fig. 2a, b). Notably, the 70% deleted sample displayed a significant reduction of Aβ accumulation which contrasted with the wide distribution of Aβ deposition found in the unedited 6-month-old *NL-G-F* mice (Fig. 2a). On the other hand, no clear differences in Aβ accumulation were observed between unedited *NL-G-F* mice and the 6% deleted sample (Fig. 2a). Consistent with these results, the deletion efficiency inversely correlated with Aβ accumulation in *NL-G-F* ΔUTR cortex (Fig. 2b), hippocampus and subcortical regions (Supplementary Fig. 2). We also checked the glial response in the *NL-G-F* ΔUTR cortex, and found a substantial reduction of the

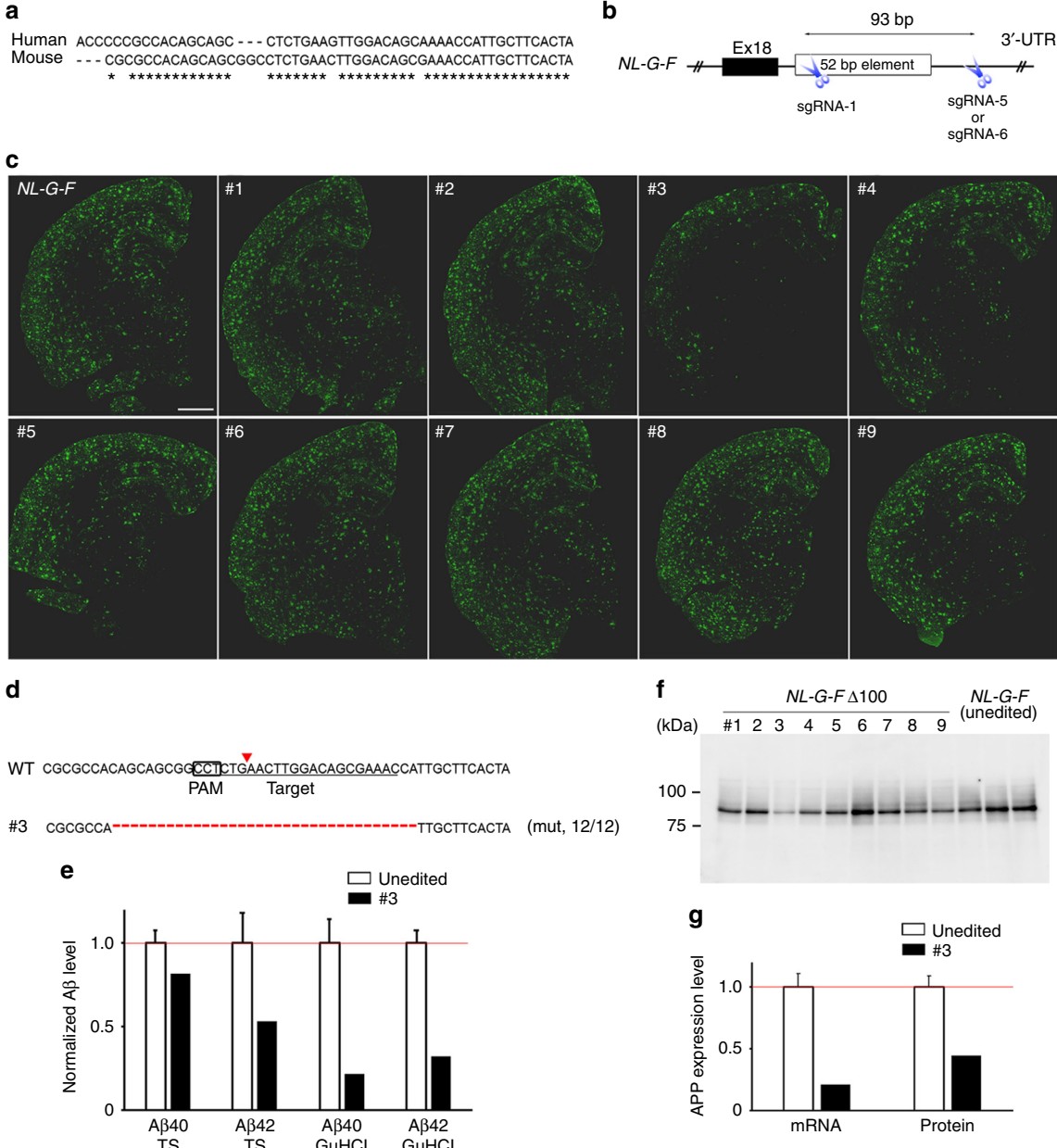

**Fig. 3** In vivo exploration of the regulatory elements on *App* 3′-UTR. **a** Sequence conservation of 52-bp regulatory element between the mouse and human. **b** Strategy for in vivo disruption of the 52-bp element by CRISPR/Cas9. Positions of targeting sgRNAs are shown by scissors. **c** Aβ pathology in 6-month-old *NL-G-F* Δ100, as well as unedited control *NL-G-F* brains. Scale bar: 1 mm. **d** Sequence analysis of founder #3 with bi-allelic 34-bp deletion. The fractions on the right indicate the mutant reads number out of total reads number. **e** Reduced Aβ levels in a 6-month-old *NL-G-F* Δ100 mouse brain (#3). Three unedited *NL-G-F* samples were used for normalization. The unedited control data represent mean ± SD. **f** A representative APP western blot image for 6-month-old *NL-G-F* Δ100 mouse brains. **g** Reduced APP expression at transcriptional and translational levels in a *NL-G-F* Δ100 mouse brain (#3). Three unedited *NL-G-F* samples were used for normalization. The unedited control data represent mean ± SD

GFAP-positive area in highly edited *NL-G-F* ΔUTR cortex (Supplementary Fig. 3). To clarify the underlying mechanisms of the observed reduction in Aβ deposition, we performed real-time PCR analyses to quantify total RNA in samples from 6-month-old *NL-G-F* ΔUTR cortexes. GAPDH-normalized APP mRNA expression levels positively correlated with Aβ accumulation in *NL-G-F* ΔUTR mice (Supplementary Fig. 4). We also validated the correlation at the protein level: APP protein expression levels significantly correlated with Aβ accumulation in *NL-G-F* ΔUTR mice (Fig. 2c); although, the intensity between unedited samples were somewhat varied possibly due to the presence of

fractionation process (Supplementary Fig. 5). Furthermore, we performed quantitative Aβ ELISA, and confirmed that the deletion efficiency inversely correlated with both $A\beta_{40}$ and $A\beta_{42}$ levels in *NL-G-F* ΔUTR cortex (Fig. 2d–g). These results provide the first evidence that CRISPR/Cas9-mediated deletion of the *App* 3′-UTR in AD model mouse zygotes mitigates Aβ pathology development via reduction of APP mRNA and protein expression levels.

Further, we injected another combination of CRISPR tools, which targets ~400-bp of *App* 3′-UTR, into *NL-G-F* mouse zygotes (Fig. 2h). Simultaneous genome editing revealed that

deleted fragments of the 3′-UTR were detected in 14 founder mice (named *NL-G-F Δ400*) out of 62 live-born pups (Supplementary Fig. 6 and Supplementary Table 2). We excluded three mice with relatively large insertions or inverted sequence from the founder candidates. Consistent with the results for the *NL-G-F ΔUTR* mice, subsequent ELISA of 6-month-old *NL-G-F Δ400* cortexes showed a clear inverse correlation between Aβ levels and the deletion efficiencies (Fig. 2i). We also found the positive correlation between Aβ accumulation and APP expression (Fig. 2j). The clear negative correlation between APP mRNA expression and genome editing efficiency in wild-type Δ400 cortex further confirmed the presence of regulatory elements of APP expression (Supplementary Fig. 7).

**52-bp element is possible sequence counteracting Aβ pathology.** Previous in vitro and cell culture experiments identified several regulatory elements within the corresponding 400-bp region of the human homologue[13,14]. Of them, a 52-bp element adjacent to a stop codon on the last exon of the *App* gene seemed to play an important role in APP mRNA stabilization. Importantly, the 52-bp element is highly conserved between mice and humans (Fig. 3a), although it remains unclear whether the 52-bp element has the same role in mouse brain in vivo. To narrow down the responsible elements on the *App* 3′-UTR, we injected another combination of CRISPR tools to cut out an ~100-bp region, including the 52-bp element, into *NL-G-F* mouse zygotes (Fig. 3b). Two weeks after birth of the mice, we performed PCR-based genotyping using tail genomic DNA, and identified the founder candidates with deleted fragments (Supplementary Table 2). Based on the Sanger sequencing results, we excluded two mice with relatively large insertions from the founder candidates. We also performed PCR-based genotyping with the brain genomic DNA, and calculated the deletion efficiency (Supplementary Fig. 8). To determine the deleted sequences of nine founder mice (named *NL-G-F Δ100*), we performed TA cloning with the PCR-amplified DNA fragments derived from each founder mouse brain. We randomly picked up ~10 colonies for each founder, and then performed colony sequencing. Expectedly, ~100-bp region of *App* 3′-UTR were deleted in seven out of nine founders (Supplementary Fig. 9). Immunohistochemical analyses with anti-Aβ antibody were performed using tissue from 6-month-old *NL-G-F Δ100* mice and un-injected control *NL-G-F* mice, and their Aβ pathology was compared. Among the nine founders, one founder (#3) showed a clear reduction in Aβ plaque area (Fig. 3c). Subsequent colony sequencing of the relevant brain sample revealed that a bi-allelic 34-bp deletion within the 52-bp element had occurred in that founder (Fig. 3d). To validate the immunohistochemistry result, we used an ELISA assay to quantitatively detect the alteration of Aβ levels in the CRISPR-injected founder mice, and found a substantial reduction of cortex Aβ levels in founder #3 (Fig. 3e). Importantly, the bi-allelic 34-bp deletion resulted in a substantial but not complete reduction in APP expression (Fig. 3f, g), possibly excluding unintended side effects derived from complete APP deficiency[15,16] because heterozygous and homozygous *App* knock-out mice exhibit no and little abnormality, respectively, due to the functional redundancy of the APP family[17]. These results strongly support the previous results obtained from human cell culture experiments, and suggest that the 34-bp deletion could be used as a protective mutation to counteract the pathological changes in FAD. Although several other *NL-G-F Δ100* mice also showed decreased Aβ levels (Supplementary Fig. 10) with somewhat reduced APP expression (Supplementary Fig. 11), no negative correlation was observed between the deletion efficiency and Aβ levels or APP expression levels. However, it was difficult to conclude that the 100-bp

disruption did not affect Aβ accumulation through the regulation of APP expression due to the relatively low-deletion efficiency (10–30%) (Supplementary Fig. 8), as well as the genetic mosaicism of the *NL-G-F Δ100* mice (Supplementary Fig. 9). Further investigations using the heterozygous or homozygous mutant offspring with various deletion mutations could lead to the in vivo identification of the responsible elements.

## Discussion

With the rapid progress of sequencing technologies, genetic variations affecting the risk of AD has been directly explored in the human population. Although such random screening using the large human dataset has been successful with the identification of some critical genetic variants[3,18,19], it is still challenging to extract lower frequency mutations. In fact, the allele frequencies of the identified variants are much higher than those of all genetic variants within the 52-bp element on *App* 3′-UTR in the general population from the Exome Aggregation Consortium (ExAC)[20] and the Genome Aggregation Database (gnomAD) (Supplementary Fig. 12). Moreover, the conventional screening strategy is not applicable to the identification of variants that do not occur naturally in the human population. To complement the random screening based on a large human dataset, we utilized genome editing-based targeted screening focusing on the *App* 3′-UTR for exploration of protective deletion mutations. As shown in this study, combination of CRISPR/Cas9 technology with the relevant *App* knock-in mouse model might allow direct evaluation of the protective effects derived from deletion mutations on Aβ deposition.

A large number of clinical trials have already been undertaken or are ongoing, but to date, no effective medications are available for AD. Given that AD affects ~34 million individuals worldwide[21], there is an urgent need to develop strategies that take a new approach to preventing AD. Apart from AD, several loss-of-function variants, which occur naturally in the human population, have been identified as protective mutations against incurable conditions: e.g., a 32-bp deletion mutation on the CCR5 coding region for HIV infection[22], and a 13-bp deletion mutation on the HBG1 promoter for hemoglobinopathies[23]. Utilizing this knowledge of protective mutations, ex vivo or in vivo genome editing strategies targeting the endogenous genes are becoming possible treatment options for these diseases[24–26]. The identification of a deletion mutation on CCR5 has also resulted in the development of an effective antiretroviral antagonist-based therapeutic approach[27,28]. In a similar fashion, the identification of novel protective mutations could provide new opportunities for the development of a novel therapeutic application for AD.

## Methods

**Animal use and care.** We used 6-month-old CRISPR-injected *App* knock-in mice[7], as well as unedited control model mice (C57BL/6). Both male and female mice were used. All experimental procedures were conducted in accordance with standard guidelines for animal experiments of RIKEN Center for Brain Science.

**Preparation of Cas9 and sgRNAs.** We constructed the plasmid vector px330 (Addgene #42230) expressing single-guide RNA (sgRNA) as well as Cas9. The px330 plasmid[9] was a gift from Feng Zhang, and the sgRNA was designed in silico via the CRISPR Design tool[29] or sgRNA designer[30]. For in vitro synthesis of Cas9 mRNA, plasmid vector pCAG-T3-hCAS-pA (Addgene #48625) was linearized by SphI, then transcribed with T3 RNA polymerase (Promega, Madison, WI, USA) in the presence of Ribo m7G Cap Analog (Promega) as described previously[31]. pCAG-T3-hCAS-pA was a gift from Wataru Fujii & Kunihiko Naito. sgRNAs were synthesized in vitro as previously described[32] using a MEGAshortscript T7 kit (Thermo Fisher Scientific, Waltham, MA, USA). Synthesized RNAs were purified using a MEGAclear kit (Thermo Fisher Scientific). All sequences of oligonucleotides for sgRNA synthesis are listed in Supplementary Table 1.

**Microinjections into mouse embryos**. The Cas9 mRNA and sgRNAs were injected into C57BL/6 mouse zygotes or *NL-G-F* mouse zygotes using a micro-injection system under standard conditions (as detailed in Supplementary Table 2). The injected zygotes were cultured in culture medium at 37 °C in 5% $CO_2$ up to two-cell embryos, and then the embryos were transferred into the oviduct of the recipient mice.

**Genotyping analysis**. Genomic DNA was extracted from tail biopsies using a standard ethanol precipitation method. Genomic DNA from brain samples was isolated using TRIzol (Ambion) reagent according to the manufacturer's instructions. Genomic DNA was amplified using a specific primer set to identify founder mice with *App* 3′-UTR deletion (Supplementary Table 1). PCR products that were ~100 bp or smaller than parental band were defined as deleted fragments. Intensity of the deleted band and intact parental band were quantified by ImageJ, and the deletion efficiency of each sample was represented as the ratio of the deleted band intensity to total band intensity (i.e., deleted band intensity plus parental band intensity). Sequencing analysis was used to further determine the mouse genotype.

**Off-target analysis**. Potential off-target sites with up to 3 bp mismatches were identified in silico using COSMID[33]. Target fragments on the off-target site were amplified using specific primer sets (Supplementary Table 3), and then sequencing analyses were carried out to detect mutations.

**Immunohistochemical analysis**. Immunohistochemical staining against Aβ (N1D, 1:200)[34] and glial fibrillary acidic protein (GFAP, 1:100; MAB3402; Millipore, Darmstadt, Germany) was performed in 6-month-old *NL-G-F* mice. Paraffin-embedded mouse brain sections were immunostained as previously described[7]. Briefly, after deparaffinization, antigen retrieval was performed by autoclave treatment (121 °C for 5 min). The sections were rinsed several times, blocked for 30 min, and then incubated with primary antibody in blocking solution overnight. A biotinylated secondary antibody and tyramide signal amplification (PerkinElmer, Waltham, MA, USA) were used for detection of amyloid pathology. A secondary antibody conjugated with Alexa Fluor 488 (1:500; Invitrogen, Carlsbad, CA) was used for detection of astrocytosis. Photographic data were captured using a NanoZoomer Digital Pathology C9600 (Hamamatsu Photonics, Hamamatsu, Japan). Immunoreactive areas were quantified using Tissue Studio software (Definiens, Cambridge, MA, USA). To reduce the effect of variation between tissue sections, we used the average of data from 3 to 5 sections per mouse.

**Quantitative RT-PCR**. Total RNA was isolated from mouse cortex using TRIzol reagent (Ambion) according to the manufacturer's instructions. Total RNA (1 µg aliquots) was converted to cDNA by ReverTra Ace (Toyobo, Osaka, Japan) and oligo (dT) primer. The resultant cDNA was diluted to 1:50 and the diluted solution was used as a sample. The quantitative PCR was performed using real-time PCR master mix (Toyobo) and specific TaqMan Gene Expression Assays (Applied Biosystems, Foster City, CA, USA) for mouse APP (Mm01344172_m1) and GAPDH (Mm99999915_g1). The ABI 7900HT system (Applied Biosystems) was used for amplification and detection. Relative expression was calculated by using the comparative cycle threshold (CT) method. The relative mRNA levels were normalized to endogenous ACTB or GAPDH mRNA expression for each sample. The CT value was obtained from the amplification plot with the aid of SDS software.

**Northern blotting analysis**. Total RNA was isolated from mouse cortex using and Micro-Fast Track 2.0 kit (invitrogen). RNA samples (6 µg) were separated in 1% formaldehyde agarose gels and transferred to nylon membranes. The membranes were reacted with isotope labeled cDNA fragment generated by PCR. The membranes were reprobed with an β-actin cDNA. cDNA fragments were amplified using specific primer sets: APP forward 5′-ATGTGCAGAATGGAAAGT-3′ and APP reverse 5′-CAGCATACAAACTCTACC-3′, β-actin forward 5′-TCAT-GAAGTGTGACGTTGACATCCGT-3′ and β-actin reverse 5′-CTTA-GAAGCATTTGCGGTGCACGATG-3′.

**Western blotting analysis**. Mouse cortexes were homogenized in 50 mM Tris-HCl (pH 7.4) containing protease inhibitor cocktail complete mini (Roche Diagnostics) (homogenizing buffer). After centrifugation at 1200×*g*, the supernatant was centrifuged at 200,000×*g* for 20 min at 4 °C. The pellet fraction was collected and then solubilized with homogenizing buffer containing 1% Triton X-100 for 60 min on ice and centrifuged again at 200,000×*g* for 20 min at 4 °C. The supernatants served as protein samples for further analyses to detect APP protein expression. Protein samples (20 µg) were separated by 10% SDS-PAGE, followed by transfer to PVDF membranes. After incubating with 2% ECL blocking reagent, the membranes were incubated with the N-terminal APP antibody (1:2000; 22C11, Millipore) or the C-terminal APP antibody (1:10,000; SIGMA, St. Louis, MO, USA) at 4 °C overnight. Then, the membranes were repeatedly washed and incubated with horseradish peroxidase-conjugated IgG secondary antibodies (1:5000; GE healthcare, Little Chalfont, UK). The band intensity was determined with a densitometer (LAS4000, Fujifilm). Each set of experiments was repeated twice. Results were the

mean of the multiple experiments. APP expression levels in CRISPR-injected *NL-G-F* mouse brains were normalized with those in three or four unedited *NL-G-F* mouse brains. Normalized APP expression levels correlated well between the experiments with two distinct APP antibodies (Supplementary Fig. 13). Some of the membranes were reprobed with the Syntaxin 1B rabbit polyclonal antibody (1:10,000; Synaptic Systems, Goettingen, Germany).

**Enzyme-linked immunosorbent assay (ELISA)**. The quantitation of Aβ levels in 6-month-old *NL-G-F* mice was performed as previously described[35]. Briefly, mouse cortexes were homogenized in 50 mM Tris-HCl (pH 7.6) containing 150 mM NaCl and the protease inhibitor cocktail (TBS) with a Teflon-glass homogenizer. After centrifugation at 200,000×*g* for 20 min at 4 °C, the supernatant was collected and defined as the soluble fraction (TS fraction). The pellet was dissolved in 6 M guanidine–HCl buffer containing the protease inhibitor cocktail. The solubilized pellet was centrifuged at 200,000×*g* for 20 min at room temperature and then used as the insoluble fraction (GuHCl fraction). The amounts of $Aβ_{40}$ and $Aβ_{42}$ in each fraction were determined by Aβ ELISA kit (Wako Pure Chemicals, Osaka, Japan) according to the manufacturer's instructions. Results were the mean of duplicate determinations. Because all Aβ peptides in *NL-G-F* mouse brains contain the Arctic mutation within the peptide sequence, the levels of $Aβ_{40}$ and $Aβ_{42}$ were quantified based on standard curves using synthetic human Arctic Aβ peptides (Peptide Institute, Osaka, Japan)[7]. Aβ levels in CRISPR-injected *NL-G-F* mouse brains were normalized with those in three or four unedited *NL-G-F* mouse brains.

**Statistical analyses**. The data were first analyzed for normal distribution. When normally distributed, the variables were statistically analyzed using two-tailed Pearson's correlation coefficient. If the data did not pass normality testing, Spearman's correlation coefficient was used. All analyses were completed with Statcel 3 (add-in software for Excel, Microsoft), and values of $P < 0.05$ were considered to be statistically significant.

**Data availability**. The data related to the findings of this study are available from the corresponding author on request.

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

## Acknowledgements

We thank Yukiko Nagai for secretarial assistance, Charles Yokoyama for comments on the manuscript, and Saido laboratory members for assistance. We are grateful to the RIKEN Center for Brain Science Research Resources Division for carrying out the injections of the CRISPR/Cas9 system into mouse zygotes, as well as for technical help with DNA sequencing analyses. This work was supported financially by an Incentive Research Grant from RIKEN (K.N. and F.O.-U.), the Special Postdoctoral Researchers Program from RIKEN (K.N. and S.H.), RIKEN Center for Brain Science (T.C.S.), Brain Mapping by Integrated Neurotechnologies for Disease Studies (Brain/MINDS) from Japan Agency for Medical Research and Development (T.C.S.), and the Aging Project of RIKEN (T.C.S.).

## Author contributions

K.N. and T.C.S. conceived the study. K.N. and T.S. designed the experiments. K.N., M.T., Y.M., F.O.-U. and K.S. performed the experiments. K.N. analyzed the data. S.H. and T.S. provided biological specimens and technical assistance. K.N. and T.C.S. wrote and edited the manuscript. All authors provided feedback and agreed on the final manuscript.

## Additional information

**Competing interests:** T.S., S.H., Y.M., and T.C.S. serve as advisor, research scientist, director, and CEO, respectively, for RIKEN BIO Co. Ltd., a RIKEN venture based at RIKEN Center for Brain Science, Central Building, Wako, Japan.

