## [Peer Review File · Nature Communications]

Reviewers' Comments:

Reviewer #2:

Remarks to the Author:

In their manuscript, "Targeted screening using single App knock-in mice for protective deletion mutations against the Alzheimer's disease pathology," Nagata et al use CRISPR-Cas technology to make deletions in the 3'UTR of APP in embryos, in a human mutant APP knock-in mouse line previously described by the same lab. They show that the efficiency of the targeting is correlated with lowering of plaque burden in the 6-month brain, with deletions introduced of 400-900 bp and then of <100 bp in the final figure. The conclusion of this study is that deletion of regions of the 3'UTR of APP affects APP levels, and therefore amyloid beta levels. Although it was already known that these sequences in the 3'UTR of APP regulate APP expression levels in cell lines (as cited by the authors in the manuscript), this is the first time that the relationship between the 3'UTR and APP expression was shown in vivo. This study is valuable to the field, but certain points must be addressed to make this suitable for publication.

- 1) The authors show abeta levels using ELISA for the final targeting, but not for the majority of data interrogating the 400-900 bp deletions. ELISA is a quantitative measure that should be shown for all data. The methods state the ELISA data were acquired for both the soluble and insoluble fractions – both should be shown. It is not clear what is shown in figure 3. The correlation coefficient to genome editing efficiency should be shown for both fractions.

- 2) The methodology used for quantifying % cleavage of targeting is not clear and needs to be clearly delineated since this is a major aspect of the study. The method used is unclear in both the methods, main text, and figure legends.

- 3) The figure legends need more information:
 - a) 3E – is this from a WB? There should be error bars for combining data from more than one well to show variability in the measure, and error around the red mean line for controls, to show variability across the different (untargeted) controls – perhaps could use gray shading. Also should show representative WB in main figure.
 - b) 2C – this is WB data – for N or C terminal antibody?
 - c) 3F – is this qPCR data? Normalized to what? Again, add error around red mean line for controls

4) Figure 2E,F has a big problem – the text states an n=14, but there are 13 dots in E and only 8 in F. Also, the genome editing efficiency doesn't match for the dots in E and F!! There must have been an error in the analysis or graphing.

5) Protein-level data should be shown for delta 400 animals in Figure 2.

6) The authors Incorrectly refer to Sup. Fig 4 when it should state Sup. Fig 5. The methods used to generate Sup. Fig.5 and the interpretation of these results relative to Supp. Fig 4 (WB) need to be better explained in the text.

Other points:

The title is somewhat confusing – consider rewording. Perhaps “Use of single App knock-in mice for protective deletion mutations against the Alzheimer's disease pathology”.

In abstract, should specify the type of APP knock in mouse line used (human, mutant).

The first sentence of the main text is not accurate. AD is not pathologically characterized by plaque deposition alone. High plaque burden can occur in non-AD individuals as well.

Incorrect reference to sup fig. 3 as being the off target genes investigated. Need to list these, and the justification for these 7 – are these the 7 with the highest identity to gRNAs?

Reviewer #3:

Remarks to the Author:

I reviewed the original manuscript for Nature medicine and thought that it was an interesting paper with important implications for future therapeutic strategies. This revised version for Nature communication is much improved and addresses my previous concerns.

Responses to reviewers

<Reviewer #2>

Comments (Major points)

Comment 1: The authors show abeta levels using ELISA for the final targeting, but not for the majority of data interrogating the 400-900 bp deletions. ELISA is a quantitative measure that should be shown for all data. The methods state the ELISA data were acquired for both the soluble and insoluble fractions – both should be shown. It is not clear what is shown in figure 3. The correlation coefficient to genome editing efficiency should be shown for both fractions.

Response: We appreciate the reviewer's suggestion. We newly performed ELISA to quantitatively measure A β 40 and A β 42 levels in both NL-G-F Δ 400 mouse brains (Fig. 2i) and NL-G-F Δ UTR mouse brains (Fig. 2d-g). To clearly display what is shown in figure 3, we have divided the previous version into two different parts: scatter plots showing the relationship between the genome editing efficiency and A β levels or APP expression levels in eight NL-G-F Δ 100 mice with heterozygous deletions (Supplementary Fig. 9) and bar graphs showing reduced A β levels and APP expression levels in a NL-G-F mouse with biallelic 34 bp deletions (Fig. 3e, g). We believe that the new version is easier to understand than previous ones. All ELISA data have been shown for both the soluble and insoluble fractions.

Comment 2: The methodology used for quantifying % cleavage of targeting is not clear and needs to be clearly delineated since this is a major aspect of the study. The method used is unclear in both the methods, main text, and figure legends.

Response: We appreciate the reviewer's suggestion. We realized that the previous version was confusing for the following reasons. First, our definition of deleted band was ambiguous. Second, the explanation for the quantifying method was often too simple. Third, electrophoresis data used for calculation was shown only in the NL-G-F Δ UTR mouse brains. In this study, deletion efficiency was calculated as previous studies (e.g. Ousterout et al., Nat Commun. 2015, 6:6244). After the amplification of genomic DNA with specific primer sets, parental band and deleted fragments can be clearly distinguished by agarose electrophoresis. Then, we measure the intensity of the deleted bands as well as parental band using ImageJ. The deletion efficiency of each sample was represented as the ratio of the deleted band intensity to total band intensity (i.e. cleaved band intensity plus parental band intensity). We have accordingly modified the methods (p.21, line 11–14), main text (p.6, line 6–8), a figure (Fig. 1c) and the figure legend (p.17, line 5–8). Also, we have newly included the electrophoresis data from NL-G-F Δ 400 mouse brains (Supplemental Fig.5) and NL-G-F Δ 100 mouse brains (Supplemental Fig.7).

Comment 3: The figure legends need more information:

a) 3E – is this from a WB? There should be error bars for combining data from more than one well to show variability in the measure, and error around the red mean line for controls, to show variability across the different (untargeted) controls – perhaps could use gray shading. Also should show representative WB in main figure.

b) 2C – this is WB data – for N or C terminal antibody?

c) 3F – is this qPCR data? Normalized to what? Again, add error around red mean line for controls

Response:

a) 3E was data from multiple WB. We have added the information of technical replicate in method section (p.24, line 9–10). In accordance with the reviewer's advice, we have plotted the standard deviation for controls to show variability in the measurement for A β levels (Fig. 3e) as well as APP expression levels (Fig. 3g). Also, we have added the representative WB data in main figure (Fig. 3f).

b) This is WB data for N terminal antibody. The information has been added in the figure legend.

c) Fig. 3F is qPCR data normalized to GAPDH mRNA.

Comment 4: Figure 2E,F has a big problem – the text states an n=14, but there are 13 dots in E and only 8 in F. Also, the genome editing efficiency doesn't match for the dots in E and F!! There must have been an error in the analysis or graphing.

Response: As we stated in the text, we plotted 14 dots in previous Figure 2E. But, two overlapping dots might have been seen as one dot. Whereas previous figure 2E indicated the result of NL-G-F Δ 400, previous figure 2F represented the data for wild-type Δ 400. We apologize that we wrote a wrong strain name (NL-G-F Δ 400) in the figure legend. The number of dots were correct. To avoid confusion of the readers, the data for wild-type Δ 400 have been moved to supplementary Figure (Supplementary Fig.6)

Comment 5: Protein-level data should be shown for delta 400 animals in Figure 2.

Response: We appreciate the reviewer's suggestion. We have added a scatter plot showing the correlation between A β plaque area and APP protein level in NL-G-F Δ 400 mouse brains (Fig. 2j).

Comment 6: The authors Incorrectly refer to Sup. Fig 4 when it should state Sup. Fig 5. The methods used to generate Sup. Fig.5 and the interpretation of these results relative to Supp. Fig 4 (WB) need to be better explained in the text.

Response: We appreciate the reviewer pointing this out and have corrected the figure number. We have added the methods used to generate previous Sup. Fig 5 in main text (p.9, line 8-12) and the figure legend. We have also included the interpretation of Sup. Fig 4. (p. 7, line 11-14).

Comment (other points):

Comment 7: The title is somewhat confusing – consider rewording. Perhaps “Use of single App knock-in mice for protective deletion mutations against the Alzheimer's disease pathology”.

Response: In accordance with the reviewer's advice, we have revised the title to “Application of single App knock-in mice for protective deletion mutations against the Alzheimer's disease pathology”.

Comment 8: In abstract, should specify the type of APP knock in mouse line used (human, mutant).

Response: We have specified the type of App knock-in mouse line used in this study.

Comment 9: The first sentence of the main text is not accurate. AD is not pathologically characterized by plaque deposition alone. High plaque burden can occur in non-AD individuals as well.

Response: We have revised the first sentence to “Alzheimer's disease (AD) is a progressive neurodegenerative disorder, which is pathologically characterized by deposition of amyloid beta peptide (A β) as well as hyperphosphorylated tau aggregation”.

Comment 10: Incorrect reference to sup fig. 3 as being the off target genes investigated. Need to list these, and the justification for these 7 – are these the 7 with the highest identity to gRNAs?

Response: Please refer to Supplementary Table 3 containing the list of seven potential off-target sites. Yes, these seven are the sites with highest identity to the target sequences.

Reviewers' Comments:

Reviewer #2:

Remarks to the Author:

The authors have responded to my critiques in an acceptable manner.

Responses to reviewers

<Reviewer #2>

Comments

Comment 1: The authors have responded to my critiques in an acceptable manner.

Response: We really appreciate the reviewer's comments useful for strengthening our manuscript.